# Adverse Effects of Aβ_1-42_ Oligomers: Impaired Contextual Memory and Altered Intrinsic Properties of CA1 Pyramidal Neurons

**DOI:** 10.3390/biom14111425

**Published:** 2024-11-08

**Authors:** Hiroyuki Kida, Itsuki Kanehisa, Masahiko Kurose, Junko Ishikawa, Yuya Sakimoto, Ryoichi Kimura, Dai Mitsushima

**Affiliations:** 1Department of Physiology, Yamaguchi University Graduate School of Medicine, Yamaguchi 755-8505, Japan; minkaung.mlm@gmail.com (M.-K.-W.-M.); h-kida@yamaguchi-u.ac.jp (H.K.); i029eb@yamaguchi-u.ac.jp (I.K.); b030eb@yamaguchi-u.ac.jp (M.K.); junko-lc@yamaguchi-u.ac.jp (J.I.); ysaki@yamaguchi-u.ac.jp (Y.S.); pawmto@gmail.com (P.-M.-T.-O.); 2Center for Liberal Arts and Sciences, Sanyo-Onoda City University, Sanyo-Onoda 756-0884, Yamaguchi, Japan; rkimura@rs.socu.ac.jp; 3The Research Institute for Time Studies, Yamaguchi University, Yamaguchi 753-8511, Japan

**Keywords:** Aβ_1-42_ oligomers, contextual memory, intrinsic properties, neuronal hyperexcitability, riluzole, lecanemab

## Abstract

Aβ_1-42_ (amyloid beta) oligomers, the major neurotoxic culprits in Alzheimer’s disease, initiate early pathophysiological events, including neuronal hyperactivity, that underlie aberrant network activity and cognitive impairment. Although several synaptotoxic effects have been extensively studied, neuronal hyperexcitability, which may also contribute to cognitive deficits, is not fully understood. Here, we found several adverse effects of in vivo injection of Aβ_1-42_ oligomers on contextual memory and intrinsic properties of CA1 pyramidal neurons. Male rats underwent behavioral and electrophysiological studies 1 week after microinjections into the dorsal CA1 region, followed by histological analysis. After 1 week, Aβ_1-42_ oligomers impaired contextual learning without affecting basic physiological functions and triggered training-induced neuronal excitability. Furthermore, riluzole, a persistent sodium current (*I*_NaP_) blocker, dose-dependently reduced Aβ_1-42_ oligomer-induced hyperexcitability. Congo red staining, which detects insoluble amyloid deposits, further identified labeling of CA1 pyramidal neurons while immunohistochemistry with lecanemab, which detects soluble Aβ oligomers, revealed immunoreactivity of both pyramidal and non-pyramidal cells in the target area. Therefore, our study suggests that a single injection of Aβ_1-42_ oligomers resulted in contextual memory deficits along with concomitant neuronal hyperexcitability and amyloid deposition in the CA1 region after 1 week.

## 1. Introduction

The pathological hallmarks of Alzheimer’s disease (AD) include amyloid plaques composed of amyloid-β (Aβ) peptides and neurofibrillary tangles containing tau aggregates [1]. Despite synergistic effects between the two pathologies [2], the aggregation and accumulation of Aβ peptides are thought to trigger AD [1]. Aβ deposition precedes tau tangle pathology [3,4], whereas tau has less influence on the development or progression of Aβ pathology [5]. This is supported by the clinical findings of AD patients who progress sequentially from Aβ to tauopathy [6], suggesting a role upstream of tauopathy, as shown in the amyloid cascade hypothesis [5].

The transmembrane protein, amyloid precursor protein (APP), is processed by sequential enzymatic cleavage of the beta [7] and gamma secretase enzyme complexes [8] to release Aβ species, terminating at amino acid residues 40 (Aβ_1-40_) and 42 (Aβ_1-42_). Although the Aβ_1-42_ peptide is a minor component of both deposited and secreted Aβ species in AD brain, it has been found to accumulate in the senile plaques of all types [9] and to be more amyloidogenic than the Aβ_1-40_ peptide [10] as the two additional hydrophobic residues at the C-terminus facilitate the rapid formation of more stable β-sheet structures by Aβ_1-42_ [11]. Because of the known association between the propensity to form higher-order assemblies and neurotoxicity [11], knowledge of the structural toxicity of Aβ_1-42_ peptides would greatly facilitate the knowledge of the causative role of the Aβ_1-42_ peptide in AD and the development of therapeutic agents.

The amyloid cascade hypothesis initially proposed the link between cognitive deficits and amyloid plaques in AD. Subsequently, several lines of evidence have linked the earliest amyloid toxicities to soluble Aβ species, although the Aβ peptide can exist in multiple forms, including monomers, oligomers, protofibrils and fibrils [12]. An increasing number of studies support the role of oligomeric Aβ as being the critical initiator of toxicity in the pathogenesis of AD [1]. For example, cognitive impairment and synaptic dysfunction have been shown to precede the formation of amyloid plaques [13,14], and these phenomena correlated and coincided with increases in Aβ_1-42_ oligomer levels [15]. Aβ_1-42_ oligomers have been isolated from parenchymal and vascular brain deposits as well as CSF of AD patients and discovered to be elevated in a region-specific manner [16,17,18]. Furthermore, Aβ_1-42_ oligomers appear sequentially in the brain regions associated with cognitive function, including the entorhinal cortex and hippocampus [19]. Overall, these data highlight that Aβ_1-42_ oligomers are the toxic culprit species that play a central role in the aetiology of AD [20].

The hippocampus has the ability to form episodic memories related to past events or experiences through the association of spatial, temporal, and novel information [21]. The CA1 subregion, an output structure of the hippocampus, is involved in the temporal processing of information to provide spatial or non-spatial context [22] and is important for the formation of spatial working memory [23] and spatial memory [24], as well as contextual memory [25,26] and recognition memory [27]. Interestingly, CA1 was selectively affected in the early stage of AD, which is characterized by degenerative anatomical and functional consequences with differential vulnerability along the radial, transverse, and longitudinal axes [28]. Although the effects of Aβ_1-42_ oligomers on neuronal intrinsic properties have been studied [29,30,31], synergistic effects between training and Aβ_1-42_ oligomers in vivo were completely unknown. Here, we found synergistic hyperexcitability between training and Aβ_1-42_ oligomers on CA1 pyramidal neurons, together with increased membrane resistance and decreased rheobase. Furthermore, we demonstrated a dose-dependent protective effect of riluzole (RLZ) against Aβ_1-42_ oligomer-induced hyperexcitability.

## 2. Materials and Methods

### 2.1. Animals

Male Sprague–Dawley rats, aged 28–30 days (Chiyoda Kaihatsu Co., Tokyo, Japan) were housed individually in opaque plastic cages (length: 25 cm; width: 40 cm; height: 25 cm) lined with woodchip. As estrogen was found to attenuate the neurotoxic effects of Aβ, including cognitive deficit [32], we rationalized using male rats in our study to circumvent the influence of gonadal hormones on the brain and Aβ pathology. A total of 32 rats were used for behavioral analysis (Figure 1). A total of 15 rats were used for patch clamp analysis after IA training (Figure 2) and microinjected with saline (n = 7), Aβ_1-42_ (n = 4), or Aβ_42-1_ (n = 4). A total of 9 rats were used for patch clamp analysis without training (Appendix A) and microinjected with saline (n = 4) or Aβ_1-42_ (n = 5). In addition, a total of 6 rats were used for riluzole treatment (Figure 3) and microinjected with saline (n = 3) or Aβ_1-42_ (n = 3). They were provided ad libitum access to food (MF, Oriental Yeast Co., Ltd., Tokyo, Japan) and tap water and maintained at a constant temperature of 23 ± 1 °C under a 12 h light/dark cycle (lights on: 08:00 a.m. to 8:00 p.m.).

### 2.2. Preparation of Aβ_1-42_ Oligomers

Synthetic Aβ_1-42_ and Aβ_42-1_ oligomers were kindly provided by Dr. Kimura. The oligomers were prepared according to the published protocol, which demonstrated the formation of oligomeric assemblies of Aβ_1-42_ peptide [33]. Briefly, Aβ_1-42_ peptide (1 mg) was dissolved in 221.5 mL of 100% HFIP (1,1,1,3,3,3-hexafluoro-2-propanol). The peptide solution was vortexed and subsequently dried under vacuum until complete elimination of HFIP. After this, the dried peptide was re-suspended in 40 µL of 20% DMSO and gently agitated every few minutes. Then, 160 µL of PBS was further added to the preparation to make a final concentration of 1 mM and stored at 4 °C for 24 h followed by −40 °C until further use. The stock solution was further diluted in 0.9% NaCl to achieve the desired concentration in our study. The final concentration of DMSO was 0.4%.

### 2.3. Stereotaxic Surgery and Microinjection of Oligomers

The rats were anesthetized with a mixture of intraperitoneal ketamine (100 mg/kg) and xylazine (10 mg/kg) injection and placed on the stereotaxic frame. The head was then immobilized with non-rupture ear bars. After a midline incision, the scalp was opened, and bilateral craniotomies were performed using a microdrill. The stainless-steel injector connected to a 10 µL Hamilton syringe was placed above the dorsal CA1. The stereotaxic coordinates were as follows: AP: −2.9 mm posterior to bregma, ML: ±2.2 mm from midline, DV: −3.2 mm from the skull surface. The rats received bilateral injections of equal volume of either 500 µM Aβ_1-42_ oligomers or saline into the target region. The total volume of 2.4 µL (1.2 µL/side) was injected at a rate of 1 µL/5 min. After injection, the injector was left in place for the next 5 min to allow diffusion and prevent reflux of the solution and slowly withdrawn. The scalp incision was sutured, and the animals were kept warm until full recovery. The accuracy of stereotaxic injection to the target region was checked by examination of the needle tract within the brain slices.

### 2.4. Behavioral Test Battery

We designed the behavioral test battery for a comprehensive analysis of sensory/motor functions, anxiety, pain sensitivity as well as learning and memory. The sequence of the tests was optimized by choosing the least to the most stressful conditions for the rats, minimizing the interference among the tests and circumventing the influence of previous experience. The rats were subjected to the test battery 1 week after injection in the following order: open-field test, object recognition task, Y-maze task, light/dark test, social recognition task, inhibitory avoidance task, fear conditioning test, and flinch-jump test. The animals were habituated to the dimly lit room illuminated by a single light bulb positioned overhead prior to the experiments and their performance was recorded with a video camera (IXY3, Canon Inc., Tokyo, Japan). The apparatus was cleaned with 70% ethanol and air-dried between the tests, and the interval between each test was 30 min [34].

#### 2.4.1. Open-Field Test

To assess locomotor activity, the open-field test was performed in a novel empty field arena under low-light conditions. The rats were placed in the center of an open-field box (length 45 cm; width 45 cm; height 45 cm) with opaque walls and allowed to move freely for 5 min. Parameters of exploratory behavior: center time and total traveled distance were measured.

#### 2.4.2. Object Recognition Task

We used the open-field test to habituate the rats. In the sampling phase, two identical objects were placed symmetrically at the corner of the box and spaced 10 cm from the wall. The rats were put into the box without facing the objects and allowed to explore the objects for 5 min. After 30 min, one of the objects was randomly replaced with a novel object that was similar in size but different in texture, shape and color. The interaction of the rats with the two objects in the test phase was monitored for 5 min. The objects as well as their location were counterbalanced between the animals to avoid any potential bias due to their preference. To assess object recognition memory, we measured the touching time of each object over the first 3 min of the test phase. The ratio of exploration time of the novel object to total exploration time was expressed as a percentage. Touching behavior was assumed to be pointing their nose toward the object less than 2 cm or touching the object with their forelimbs for at least 1 s, while sitting on or leaning against the object was not considered [34].

#### 2.4.3. Y-Maze Task

Working memory was assessed by recording spontaneous alternation behavior, which requires the rats to remember their previous choice to select the correct arm on the next choice [35]. The Y-maze apparatus consisted of three grey, plastic arms converged in an equilateral triangular central area (MY-10, Shin factory, Fukuoka, Japan). The wall of the arms had an outside slope of 76° (height 12 cm), allowing the rats to see distal landmarks with no inter-maze cues inside. Firstly, the animals were put at the end of one arm and allowed to move freely for 5 min. After 24 h, they were placed in one arm again, and their spontaneous behavior was recorded for 5 min. Arm entry was considered to be completed when the hind paws of the rats completely crossed the arm, and alternation was defined as consecutive entries into the three arms without overlapping. By analyzing the number and sequence of arm entries, the alternation percentage was calculated as the ratio of actual to possible alternations (defined as the total number of arm entries − 2) × 100 [34,36].

#### 2.4.4. Light–Dark Box Test

A box (length 48 cm; width 20 cm; height 23 cm), which consisted of light and dark compartments separated by a sliding door (width 7 cm; height 8 cm), was used in our study. The rats were placed in the dark side and allowed to explore for 5 min. Then, the sliding door was opened, and the animals had free access to both compartments for 5 min. Time spent in the light side, as well as latency to enter and number of entries to the light side, were evaluated to assess anxiety-like behavior [37].

#### 2.4.5. Social Recognition Task

The U-field two-choice box was created by partitioning the open-field box with a wall (length 20 cm; height 45 cm) to test social recognition. Firstly, the rats were habituated to the box containing two empty cages (8 cm in diameter) for 5 min. In the sampling phase, they were allowed to explore a cage containing a social target (female; same strain and age) and an empty cage for 5 min. In the test phase, the rats were placed again to freely interact with both familiar and novel social targets for 5 min. To assess social recognition memory, we measured the time spent touching the social targets and calculated the ratio of exploration time of the novel social target to that of both targets. Touching behavior was assumed to be directing their nose toward the cage or touching the cage with their forelimbs [38].

#### 2.4.6. Inhibitory Avoidance (IA) Task

The IA task was performed according to our previous publications [26,34]. The IA apparatus was a two-chambered box consisting of an illuminated safe compartment and a dark shock compartment (length 33 cm; width 58 cm; height 33 cm) divided by a trap door. During training, the rats were placed in the lit compartment with their head facing opposite the door. After a few seconds, the door was gradually opened to allow the animals to access the dark side at their own will. The latency to enter the novel dark compartment before shock was measured (i.e., latency before IA learning). When the rats entered the dark side, the door was closed, and an electrical foot shock (1.6 mA, 2 s) was delivered via steel rods on the grid floor of the chamber. After 10 s, the animals were returned to their home cage. After 30 min, they were put back in the illuminated compartment and retrieval was performed with a criterion of 600 s. The latency to enter the previously experienced dark compartment aftershock (i.e., latency after IA learning) was measured as an index of contextual learning.

#### 2.4.7. Fear Conditioning Test

The fear conditioning paradigm was described previously [34,39]. The conditioning chamber (length 25 cm; width 31 cm; height 42 cm) consisted of Plexiglass on the top, front, and back, with 18 stainless steel bars (ϕ 4 mm; 15 mm spacing) on the bottom into which the coded shocks were delivered via a stimulator (LE100-26 Shocker, Panlab, Cornellà, Spain). During training, the rats were allowed to explore the chamber for 3 min and received an electrical foot shock as an unconditioned stimulus (0.8 mA, 2 s duration) three times with an interval of 30 s. They were allowed to recover in the chamber for 30 s and returned to their home cage. After 24 h, the rats were again placed in the conditioning chamber, and freezing was monitored every 30 s for 5 min. Freezing was defined as the cessation of all behaviors except respiration for at least 1 s [40].

#### 2.4.8. Flinch-Jump Test

To assess pain sensitivity, the flinch-jump test was performed as reported previously [34,41]. The rats were placed individually in the conditioning chamber mentioned above and habituated for 30 s. Then, foot shocks were applied in a stepwise manner (0.05 mA increments; range 0.05–0.6 mA) with an interval of 30 s, and each animal was tested only once at each intensity. The “flinch” and “jump” thresholds were defined as the lowest shock intensities that elicited a detectable response and simultaneous removal of at least three paws (including both hind paws) from the grid, respectively. The “vocalization” threshold was defined as the minimum shock intensity at which the rats produced a detectable vocalization in response to the shock.

### 2.5. Electrophysiology

For electrophysiological recording, the rats received a unilateral injection of 100 µM Aβ_1-42_ oligomers into the right dorsal CA1 region (AP: −2.9 mm posterior to bregma, ML: ±2.2 mm from midline, DV: −2.5 mm below the dura surface). After 1 week, the IA task was performed, and the rats were sacrificed 60 min after the foot-shock. After confirming there was no tail reflex, acute brain slices were prepared as described previously [26,42]. The brain was quickly perfused with ice-cold dissection buffer containing 25.0 mM NaHCO_3_, 1.25 mM NaH_2_PO_4_, 2.5 mM KCl, 0.5 mM CaCl_2_, 7.0 mM MgCl_2_, 25.0 mM glucose, 110.0 mM choline chloride, 3.10 mM pyruvic acid, 11.6 mM ascorbic acid and rapidly transferred to the chamber containing ice-cold dissection buffer, which was continuously gassed with 5%CO_2_/95%O_2_. After removal of the cerebellum and frontal parts, coronal brain slices (350 μm) were cut with a Leica vibratome (VT-1200; Leica Biosystems, Nussloch, Germany) submerged in the dissection buffer. The slices were then transferred and incubated in the storage chamber containing artificial cerebrospinal fluid (aCSF) solution (118 mM NaCl, 2.5 mM KCl, 26 mM NaHCO_3_, 1 mM NaH_2_PO_4_, 10 mM glucose, 4 mM MgCl_2_ and 4 mM CaCl_2_, pH 7.4) at 22–25 °C equilibrated with 5% CO_2_/95% O_2_ for 1 h before recording. We kept 3–4 slices from each brain and then selected 1–2 slices for a patch clamp based on the brain atlas by Paxinos and Watson [43].

### 2.6. Current Clamp Recordings

The slices were placed into the recording chamber, which was continuously perfused with aCSF solution bubbled with carbogen at 22–25 °C. The dorsal CA1 pyramidal neurons were visually identified using infra-red (IR) differential interference contrast optics. The pipettes (7–9 MΩ) were made from borosilicate glass by using a horizontal puller (Model P97; Sutter Instrument, Novato, CA, USA) and filled with an internal solution containing 30 mM K-Gluconate, 5 mM KCl, 10 mM HEPES, 2.5 mM MgCl_2_, 4mM Na_2_ATP, 0.4 mM Na_3_GTP, 10 mM Na-phosphocreatine and 0.6 mM EGTA at pH 7.25. After giga-seal formation, whole-cell configuration was made from the soma of pyramidal neurons, using an Axopatch1D amplifier (Molecular Devices Inc., San Jose, CA, USA) in the current clamp mode. The liquid junction potential was not corrected. The recordings were low-pass filtered (5 kHz), digitalized with a Digidata 1440 AD board and analyzed offline using the pCLAMP 10 software (Molecular Devices).

Resting membrane potential (RMP) was evaluated before the current injection. To study the relationship between firing frequency and current injection, a square step-current, ranging from −100 to +550 pA of 300 ms duration in 50 pA increments was injected. The number of spikes was counted and the current threshold or rheobase was determined as the minimum current intensity to evoke at least a single spike. The threshold potential of the first AP at 300 pA was derived from a point where dV/dt exceeded 10 mV/ms [29]. The intrinsic properties of the membrane: membrane capacitance (C_m_) and time constant (tau) were measured by the membrane test in the voltage-clamp mode with a holding potential of −60 mV. Membrane resistance (R_m_) was also calculated as the slope of the steady-state values of the voltage responses to a series of current steps from −100 to +100 pA with 50 pA increments per step fitted with linear regression [29]. Riluzole (A2423; TCI, Tokyo, Japan) was firstly dissolved in DMSO followed by serial dilution in 0.9% NaCl to achieve desired concentrations, and the preparation was added to the perfusate at increasing concentrations (2, 4, 10, and 20 µM) with an interval of 5 min. For experiments with riluzole, each slice was used for recording one cell, as the effect of riluzole could not be completely washed off.

### 2.7. Congo Red Staining

The rats were deeply anesthetized with an overdose of intraperitoneal ketamine (750 mg/kg)/xylazine (60 mg/kg) injection and transcardially perfused with 0.9% NaCl followed by 4% paraformaldehyde (PFA). The brains were removed, post-fixed in 4% PFA for 24 h and transferred to 10%, 20% and 30% sucrose in 4% PFA at 4 °C. The fixed brains were cut on a microtome (Retritome REM-710; Yamato, Saitama, Japan) in 40 µm coronal sections. The slices were embedded in 0.8% gelatin, mounted on the glass slides and air-dried for 1 week. Congo Red staining was performed according to the manufacturer’s instruction (Congo Red staining kit, 101641; Milipore, Sigma-Aldrich, St. Louis, MO, USA) with slight modification. Briefly, the sections were washed with distilled water for 1 min and incubated in Congo Red solution for 5 min followed by rinsing with tap water for 5 min. Next, they were treated with KOH solution for 30 s and washed with tap water. Then, they were serially dipped in 96% and 100% ethanol, each two times, cleared in xylene two times and coverslipped with mounting medium (Enthellan^®^, 107961; Merck, Darmstadt, Germany). The sections were processed at the same time using the same solutions to reduce variability in staining. The staining intensity of the pyramidal cells in the ROI (region of interest) was quantified by ImageJ version 1.54a software (National Institute of Health, https://imagej.net/ij/, accessed on 13 March 2024) and expressed as integrated density (mean grey value x area).

### 2.8. Lecanemab for Immunostaining

The detailed procedure for immunohistochemistry was described in our previous publication [26]. Briefly, post-fixed brains were frozen at −80 °C and coronal sections were sliced at 30 µm using a cryostat (Leica CM1860; Leica Biosystems, Nussloch, Germany). The free-floating sections were first washed with PBS (Na_2_HPO_4_ 1.42 g and NaCl 8.75 g in 1 L, pH 7.4) 2 times for 5 min and incubated with blocking solution (1.5% normal goat serum (NGS) with 0.1% Triton X-100 in PBS) at room temperature for 1 h. They were washed with PBS 2 times for 5 min and further incubated overnight with the primary antibody, lecanemab (1:1000; A3112, Selleck Biotechnology Ltd., Yokohama, Japan), diluted in PBS containing 1.5% NGS at 4 °C under slight agitation. Then, the antigen–antibody complex in the slices was detected by using goat anti-human IgG, conjugated to Alexa Fluor^TM^ Plus 488 (1:500; A56869, Invitrogen, Waltham, MA, USA) diluted in PBS at room temperature for 1 h in the dark. Next, after washing 3 times with PBS, they were slide-mounted and coverslipped with a fluorescent mounting medium (Dako, S3023; Agilent Technologies, Santa Clara, CA, USA). The images were visualized under a fluorescence microscope (BX51N-33-FLB2; Olympus Co., Tokyo, Japan) and acquired by a digital microscope camera (DP72; Olympus Co., Tokyo, Japan). For semi-quantitative analysis with ImageJ software, 2–3 sections with ROI below the injection site per brain (number of rats = 3–4) were included. The CA1 region containing stratum oriens, stratum pyramidale and stratum radium was consistently captured at 20× magnification with the same exposure time and acquisition parameters such as frame dimension 4140 × 3096 pixels. The area of ROI was calculated in mm^2^ and the cell density of immunopositive cells was expressed as cells/mm^2^. The cell shape was verified under direct visualization and cell numbers were counted manually. Cells with morphological characteristics highly suggestive of either astrocytes or microglial [44,45] were collectively referred to as non-pyramidal cells.

### 2.9. Statistical Analysis

Statistical analysis and graphical expression were performed using GraphPad Prism 9 software (Graph Pad Software Inc., La Jolla, CA, USA). Data were expressed as mean ± SEM. The normal distribution of data was checked by a Shapiro–Wilk test. To analyze the object recognition and social recognition tasks, we used a paired *t*-test to compare the time spent on touching the novel and familiar targets in the test phase. For the IA task and fear condition test, statistical comparisons were performed with a two-way ANOVA in which the between-group factor was the oligomers, and the within-group factor was the training. Other behavioral tests and electrophysiological parameters were analyzed by one-way ANOVA in which the between-group factor was the oligomers followed by Sidak’s post hoc multiple comparison test. The relationship between the spike numbers and current was analyzed by using a two-way repeated measures ANOVA in which the between-group factor was the oligomers, and the within-group factor was the current. For RLZ-treated cells, the data were analyzed by a two-way repeated measures ANOVA in which the between-group factor was the drug, RLZ, and the within-group factor was the current. The *F* value (analysis of variance) was described, and statistical significance was set at *p* < 0.05.

## 3. Results

### 3.1. Effects of Aβ_1-42_ Oligomers on IA Learning

To investigate contextual learning performance, rats were subjected to the IA task 1 week after bilateral injections of Aβ_1-42_ oligomers into dorsal CA1. In this learning paradigm, rats were allowed to cross from a light chamber to a dark chamber where an electric foot shock (1.6 mA, 2 s) was delivered. Half an hour after the IA task, the latency to enter the illuminated side after the shock was measured as an index of contextual learning (Figure 1A). Two-way ANOVA revealed the main effects of training (*F*_1,58_ = 67.916, *p* < 0.0001) and oligomers (*F*_2,58_ = 5.951, *p* = 0.005) with a significant interaction between them (*F*_2,58_ = 6.021, *p* = 0.004). The post hoc Sidak analysis showed that the latency to enter the dark box was significantly prolonged after training in each group (saline; *p*  < 0.0001, Aβ_1-42_; *p*  = 0.046, Aβ_42-1_; *p* = 0.0025 vs. training; Figure 1B), but was shorter in both the Aβ_1-42_- (vs. saline; *p* = 0.0002) and Aβ_42-1_-injected groups (vs. saline; *p* = 0.008) compared to the saline group. Therefore, despite their successful learning, the rats that received oligomers containing both the Aβ_1-42_ sequence and its reverse Aβ_42-1_ sequence showed impaired performance during retrieval, highlighting the similar adverse effect of oligomers on contextual memory.

**Figure 1 biomolecules-14-01425-f001:**
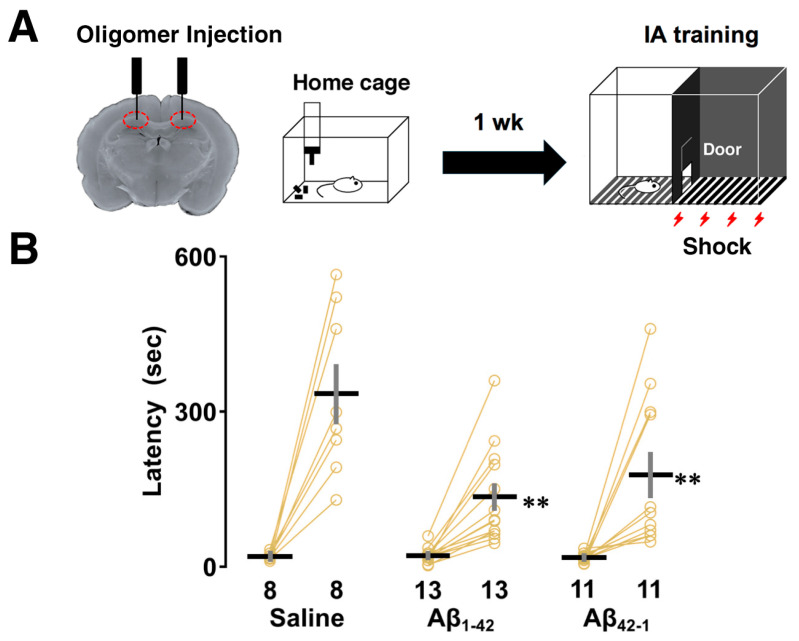
Effect of Aβ_1-42_ oligomers on the inhibitory avoidance (IA) task. (**A**) Schematic illustration of the IA task paradigm. Rats were subjected to the IA task 1 week after bilateral injections of Aβ_1-42_ oligomers into dorsal CA1. On the training day, a brief electric foot shock (1.6 mA, 2 s) was applied when the rats entered the dark compartment of the IA box. After 30 min, they were placed on the light side and the latency to enter the dark side was measured. (**B**) A shorter latency was observed in both Aβ_1-42_ and Aβ_42-1_ oligomer-injected rats. The number of rats is shown at the bottom of each bar. Data were plotted as individual points and expressed as mean ± SEM. ** *p* < 0.01 vs. saline.

### 3.2. Effects of Aβ_1-42_ Oligomers on the Intrinsic Properties After IA Learning

To assess the intrinsic properties of CA1 pyramidal neurons after contextual learning, we performed a slice patch clamp experiment using a 300 ms square current injection 1 week after unilateral injection, which did not affect learning performance (Figure 2A). Figure 2B (right panel) shows the relationship between the number of spikes and current intensity in the saline, Aβ_1-42_, and Aβ_42-1_ injected groups after the IA task. A two-way repeated measures ANOVA showed the main effects of oligomers (*F*_2,781_ = 52.224, *p* < 0.0001) and current (*F*_10,781_ = 32.036, *p* < 0.0001) with a significant interaction between them (*F*_20,781_ = 2.586, *p* = 0.0002). The post hoc Sidak analysis showed that the number of evoked spikes was increased in the Aβ_1-42_ group (vs. saline + IA; *p* < 0.0001), suggesting that cells in the Aβ_1-42_ group were more excitable after contextual learning. We also confirmed that there was no significant difference in spike counts between the saline and Aβ_1-42_ groups without the IA task (Aβ_1-42_ vs. saline; *p* = 0.678, Appendix A), highlighting the role of training-dependent changes in CA1 neuronal excitability.

Several changes in intrinsic membrane properties were observed in CA1 pyramidal neurons after the IA task. First, one-way ANOVA followed by post hoc comparison revealed that membrane capacitance was not altered (*F*_2,71_ = 1.082, *p* = 0.345; Aβ_1-42_ + IA vs. saline + IA, *p* = 0.275; Figure 2C). A significant effect of oligomers was observed in R_m_ (One-way ANOVA: F_2,71_ = 25.058, *p* < 0.0001), where neurons in both Aβ_1-42_ (vs. saline + IA; *p* < 0.0001; Figure 2D) and Aβ_42-1_ (vs. saline + IA; *p* < 0.0001) groups had higher membrane resistance after IA learning. There was no significant difference in the membrane time constant (*F*_2,71_ = 0.741, *p* = 0.481; Aβ_1-42_ + IA vs. saline + IA, *p* = 0.8411; Figure 2E). With respect to RMP, a significant effect of oligomers was found (one-way ANOVA: *F*_2,71_ = 15.949, *p* < 0.0001; Figure 2F) with the Aβ_42-1_ group having a more depolarized RMP compared to both saline (vs. saline + IA; *p* < 0.0001) and Aβ_1-42_ groups (vs. Aβ_1-42_ + IA; *p* = 0.0002), whereas RMP was not affected in the Aβ_1-42_ group (saline + IA vs. Aβ_1-42_ + IA; *p* = 0.492). 

**Figure 2 biomolecules-14-01425-f002:**
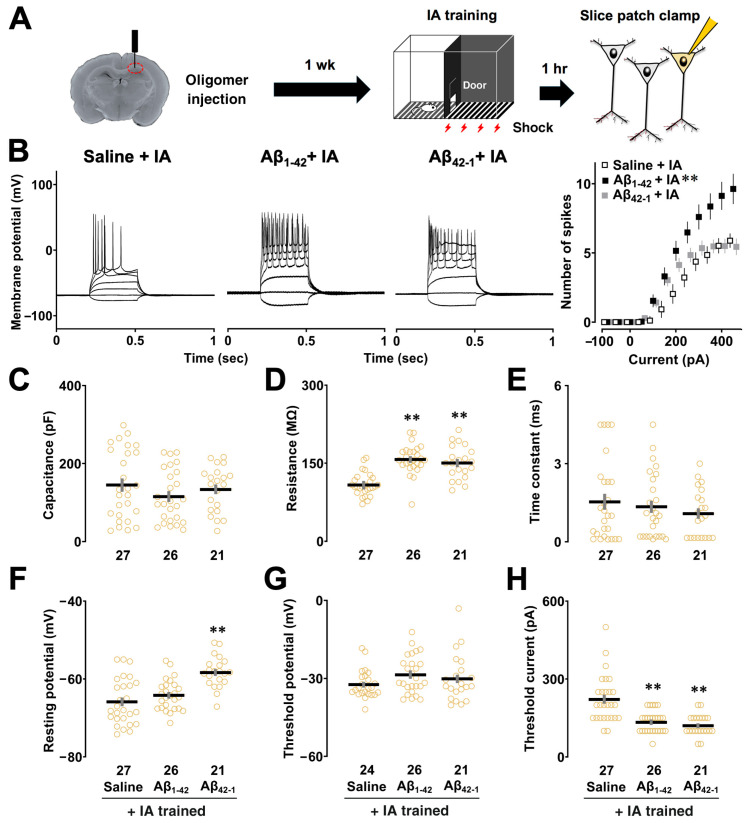
Effect of Aβ_1-42_ oligomers on the intrinsic properties of CA1 pyramidal neurons. (**A**) Schematic illustration of the slice patch clamp experiment. To assess the intrinsic properties of CA1 pyramidal neurons 1 h after IA learning, a current clamp experiment was performed 1 week after unilateral injection of Aβ_1-42_ oligomers. (**B**) Representative traces of action potentials elicited by current injections (left panel), and input–output relationship in saline, Aβ_1-42,_ and Aβ_42-1_ oligomer-injected rats with the IA task (right panel). (**C**) Membrane capacitance, C_m_. (**D**) Membrane resistance, R_m_. (**E**) Membrane time constant, tau. (**F**) Resting membrane potential, RMP. (**G**) Threshold potential. (**H**) Threshold current or rheobase of the saline, Aβ_1-42_ and Aβ_42-1_ oligomer-injected rats with the IA task. The number of cells is shown at the bottom of each bar. Data were plotted as individual points and expressed as mean ± SEM. ** *p* < 0.01 vs. saline + IA.

No significant difference was found in the threshold potential subtracted from the dV/dt of the first spike elicited at 300 pA (One-way ANOVA: *F*_2,68_ = 1.032, *p* = 0.362; Aβ_1-42_ + IA vs. saline + IA, *p* = 0.963; Figure 2G). However, one-way ANOVA showed a significant main effect associated with rheobase (*F*_2,71_ = 18.077, *p* < 0.0001), as CA1 pyramidal neurons in both Aβ_1-42_ (vs. saline + IA; *p* < 0.0001; Figure 2H) and Aβ_42-1_ groups (vs. saline + IA; *p* < 0.0001) required less current injection to evoke one AP. We found that R_m_ of the Aβ_1-42_-injected group was elevated without the IA task, although other intrinsic properties of CA1 pyramidal neurons were not altered by Aβ_1-42_ oligomers (Appendix A), suggesting that learning-dependent changes in the intrinsic properties account for the electrophysiological features of Aβ_1-42_ oligomer-induced neuronal hyperexcitability after the IA task.

To determine whether persistent sodium current (*I*_NaP_) contributes to Aβ_1-42_ oligomer-induced hyperexcitability, RLZ was bath applied to aCSF at increasing concentrations (2, 4, 10 and 20 µM) (Figure 3A,C). A two-way repeated measures ANOVA revealed the main effects of RLZ (*F*_4,280_ = 73.301; *p* < 0.0001) and current (*F*_13,280_ = 114.494; *p* < 0.0001) with a significant interaction between them (*F*_52,280_ = 3.691; *p* < 0.0001) in the Aβ_1-42_ group after the IA task. The post hoc Sidak analysis showed that spikes were less evoked after the addition of 4, 10 and 20 µM RLZ to CA1 pyramidal neurons (Figure 3B), and this inhibitory effect was enhanced with increasing concentrations (2 µM vs. 4 µM; *p* = 0.0009, 4 µM vs. 10 µM; *p* = 0.0001, 10 µM vs. 20 µM; *p* = 0.0039). A similar finding was also observed in the saline group in the IA task. A two-way repeated measures ANOVA showed the main effects of RLZ (*F*_4,420_ = 35.010, *p* < 0.0001) and current (*F*_13,420_ = 93.635, *p* < 0.0001) with a significant interaction between the two factors (*F*_52,420_ = 1.994, *p* = 0.0001), and the effect was significant after the application of 4, 10 and 20 µM RLZ (Figure 3D). Taken together, these results indicate that RLZ can inhibit Aβ_1-42_ oligomer-induced CA1 neuronal hyperexcitability in a concentration-dependent manner.

**Figure 3 biomolecules-14-01425-f003:**
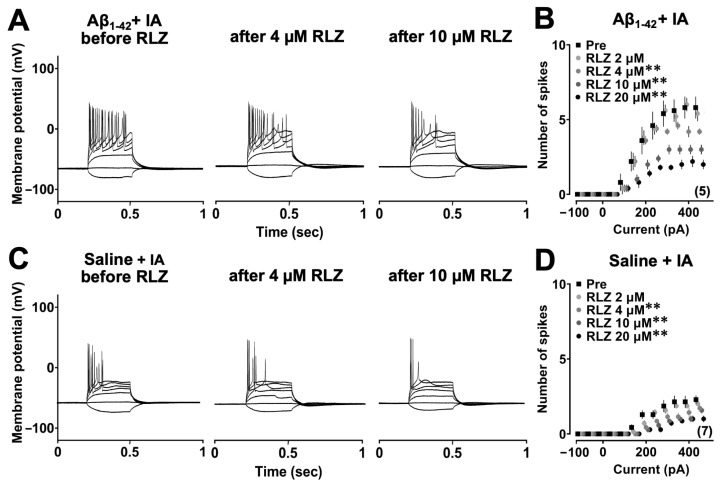
Effect of riluzole on the Aβ_1-42_ oligomer-induced hyperexcitability of CA1 pyramidal neurons. (**A**,**C**) Example traces of action potentials induced by current injections before and after riluzole in the Aβ_1-42_ oligomer-injected and saline-injected rats with IA. (**B**,**D**) Input–output relationships in the Aβ_1-42_ oligomer and saline-injected rats before and after bath application of different concentrations of riluzole. The number of cells is shown in the graph. Data were expressed as mean ± SEM. ** *p* < 0.01 vs. pre.

### 3.3. Effects of Aβ_1-42_ Oligomers on Sensory/Motor Functions, and Emotional State

Because oligomers could affect basic physiological functions, which may influence contextual learning performance, the open field, light/dark box, and flinch-jump tests were performed after 1 week of bilateral injections to rule out these possibilities (Figure 4A).

The open-field test was used to determine whether the oligomers would affect the exploratory behavior of the animals. The one-way ANOVA showed no significant main effect of oligomers on center time (*F*_2,29_ = 1.891, *p* = 0.169) and distance traveled (*F*_2,29_ = 1.297, *p* = 0.289), suggesting that the motor performance and emotional state of the animals did not differ (Figure 4B). In the light/dark test, the one-way ANOVA revealed that the main effect of oligomers was not significant for the following parameters: time spent in the light box (*F*_2,29_ = 0.522, *p* = 0.599), latency (*F*_2,29_ = 0.470, *p* = 0.630), and number of visits (*F*_2,29_ = 0.156, *p* = 0.857) to the light box, highlighting no detectable change in anxiety levels (Figure 4C). Next, the one-way ANOVA showed that there was no significant main effect associated with the current thresholds of flinch (*F*_2,29_ = 1.212, *p* = 0.312), vocalization (*F*_2,28_ = 0.197, *p* = 0.823), and jump (*F*_2,29_ = 2.868, *p* = 0.073) in the flinch-jump test, suggesting the rats had similar pain sensitivity (Figure 4D). Overall, Aβ_1-42_ oligomers did not affect sensory/motor functions, pain sensitivity, and emotional state.

### 3.4. Effects of Aβ_1-42_ Oligomers on Other Hippocampus-Dependent Tasks

We also examined the effects of oligomers on other hippocampus-dependent tasks. First, although two-way ANOVA failed to show a significant effect of oligomers (*F*_2,58_ = 3.156, *p* = 0.050), the main effect of training (*F*_1,58_ = 126.132, *p* < 0.0001) was observed with a significant interaction between training and oligomers (*F*_2,58_ = 3.271, *p* = 0.045) in the fear conditioning test. In addition, the post hoc Sidak analysis showed that freezing time was prolonged in all groups after training when re-exposed to the same context (saline; *p* < 0.0001, Aβ_1-42_; *p* < 0.0001, Aβ_42-1_; *p* < 0.0001 vs. training; Figure 5A), but the Aβ_1-42_ injected rats showed less freezing compared to the control (vs. saline; *p* = 0.037).

Next, to assess recognition memory, object recognition and social recognition tasks were administered in our test battery. In the object recognition task, the percentage of time spent on the novel object was increased in both the saline (*t* = 4.404, *p* = 0.003, paired *t*-test; Figure 5B) and Aβ_42-1_ groups (*t* = 5.291, *p* = 0.0004, paired *t*-test), but not in the Aβ_1-42_ group (*t* = 1.108, *p* = 0.290, paired *t*-test) during the test phase. In addition, one-way ANOVA showed a significant effect of oligomers (*F*_2,29_ = 4.714, *p* = 0.017), and touch time to a novel object was significantly reduced in the Aβ_1-42_ injected group (vs. saline; *p* = 0.044) in the post hoc Sidak analysis. This difference was not due to preference, as all groups showed no left/right preference in the sampling phase (Appendix A) and similar levels of exploration in the test phase (Appendix A). In the social recognition task, the percentage of social interaction time with the novel target was increased in both the saline (*t* = 4.746, *p* = 0.002, paired *t*-test; Figure 5C) and Aβ_42-1_ groups (*t* = 4.310, *p* = 0.002, paired *t*-test), but not in the Aβ_1-42_ group (*t* = 0.811, *p* = 0.434, paired t-test) during the test phase. One-way ANOVA revealed no main effect of oligomers on social interaction time (%) with the novel target (*F*_2,29_ = 2.680, *p* = 0.086), although the Aβ_1-42_-injected group showed a reduced tendency to spend time with the novel target that was not statistically significant (vs. saline; *p* = 0.192). Total exploration time of both social targets in the test phase was comparable in all groups (Appendix A). Spatial working memory was assessed by using the Y-maze task. The one-way ANOVA suggested no significant main effect of oligomers on alternation rate (*F*_2,29_ = 1.123, *p* = 0.339; Figure 5D) and number of arm entries (*F*_2,29_ = 3.079, *p* = 0.061; Figure 5E), although both parameters tended to decrease in the Aβ_1-42_ group.

### 3.5. Aβ Deposition in the CA1 Region

Congo red staining was used to determine whether injection of Aβ_1-42_ oligomers would lead to Aβ deposition (Figure 6A). A previous study suggested that a conformational change to the β-sheet structure is the first step in the aggregation process, and the Aβ_1-42_ peptide contains amino acid residues 16–20 (KLVFF) that constrain the monomer into the compact amyloid fold [46]. Congo red staining is known to detect compacted β-sheet amyloid and to label both parenchymal and vascular amyloid deposits in the AD brain [47]. One-way ANOVA showed a significant effect of oligomers on Congo red staining (*F*_2,1778_ = 402.214, *p* < 0.0001) and a post hoc Sidak comparison revealed that the integrated density of CA1 pyramidal neurons was higher in both the Aβ_1-42_ (vs. saline; *p* < 0.0001; Figure 6B) and Aβ_42-1_ (vs. saline; *p* < 0.0001) groups. In addition, the staining intensity of the Aβ_1-42_ group was also more intense than that of the Aβ_42-1_ group (Aβ_1-42_ vs. Aβ_42-1_; *p* < 0.0001).

Next, to validate the location and extent of distribution of Aβ_1-42_ oligomers in the CA1 region (Figure 6C), immunohistochemistry was performed using the anti-Aβ antibody lecanemab. Based on recent reports, lecanemab, which binds to amino acid residues (1–16) of the Aβ peptide, was found to be specific for soluble Aβ_1-42_ oligomers. In our experiments, the main effect of oligomers was significant (one-way ANOVA: F_2,27_ = 27.351, *p* < 0.0001; Figure 6D) for the Aβ_42-1_ groups (vs. Aβ_42-1_; *p* < 0.0001), whereas immunolabeling was insignificant in the latter groups (saline vs. Aβ_42-1_; *p* = 0.617). We also observed immunoreactivity along the injection track (Figure 6E, left panel) and abundant positive non-pyramidal cells (Figure 6E, right panel) along the radial axis of the CA1 region. A one-way ANOVA followed by a post hoc Sidak analysis revealed that the Aβ_1-42_ group had more lecanemab-positive non-pyramidal cells in the stratum pyramidale (vs. saline; *p* < 0.0001; Figure 6F), stratum oriens (vs. saline; *p* < 0.0001) and stratum radiatum (vs. saline; *p* < 0.0001) of the CA1 region, whereas this increase in cell density was significant only in the stratum oriens of the Aβ_42-1_ group (vs. saline; *p* = 0.0176). Thus, Congo red staining and lecanemab immunolabeling confirmed Aβ amyloid deposition along with oligomer-positive non-pyramidal cells in the CA1 region of the Aβ_1-42_ injected group.

## 4. Discussion

### 4.1. Aβ_1-42_ Oligomers Impaired CA1-Dependent IA Learning

The hippocampus provoked a particular interest in the injection model of AD [19], and its subregion CA1 is initially affected in Alzheimer’s disease [28]. CA1, a primary output to extrahippocampal circuits, integrates contextual [25] and spatial information [27], leading to the formation of episodic memory. In our study, Aβ_1-42_ oligomers impaired contextual learning 1 week after injection into CA. As contextual memory formation is dependent on the integrity of the hippocampus [25], our data suggest that Aβ_1-42_ oligomers triggered specific impairment of contextual memory related to the CA1 region.

Unexpectedly, this deficit was observed along with increased membrane resistance and amyloid deposition as evidenced by Congo red staining in both Aβ_1-42_- and Aβ_42-1_-injected rats. These findings have highlighted the possibility that amino acids in different sequences may fold into a structure that shares convergent mechanisms, and the affinity for Congo red further indicates the propensity of the reverse sequence Aβ_42-1_ to form aggregates [20]. Interestingly, one study found that both oligomers could assembly into β-sheet structure in which Aβ_1-42_ shows gradual conformational changes and the Aβ_42-1_ peptide forms fibrillar structures less ordered in appearance compared to the Aβ_1-42_ peptide [48]. This was consistent with our findings that Congo red staining of CA1 pyramidal neurons was more robust in the Aβ_1-42_ group compared to the Aβ_42-1_ control.

The deleterious effects of Aβ peptides have been shown to be specifically related to their amyloidogenicity, which in turn depends on the amino acid sequence [49]. This may be partly explained by the analysis of the WALTZ algorithm, in which we found that the two regions of the Aβ_1-42_ peptide between residues 16–21 and 37–42 are amyloidogenic, whereas a region of the Aβ_42-1_ peptide between residues 8 and 13 has similar amyloidogenic potential (Appendix A). However, the Aβ_42-1_ peptide was predicted to form more β-strands than the Aβ_1-42_ peptide based on the Phyre2 web portal (Appendix A). Furthermore, consistent with our interest, the Aβ_1-42_ peptide appears to interact directly [50] or indirectly [51] with the bilayer membrane, thereby disrupting mechanical and biophysical functions that underlie the physiological properties of the neurons. Thus, the Aβ_1-42_-induced deficit may be triggered by a direct effect of oligomers on the cell membrane of CA1 pyramidal neurons.

### 4.2. Aβ_1-42_ Oligomers Induced Neuronal Hyperexcitability After IA Learning

Several preclinical and clinical studies have suggested that neuronal hyperexcitability is strongly associated with cognitive impairment, probably by adversely affecting the intrinsic properties [52,53]. To address this issue, we performed an IA task followed by an in vitro slice patch clamp 1 week after unilateral injection, which did not affect learning performance. Membrane resistance was higher in both groups, whereas rheobase was specifically lower in the Aβ_1-42_-injected group with concomitant increased spike numbers, highlighting specific Aβ_1-42_ oligomer-induced neuronal hyperexcitability after contextual learning.

Consistent with our data, several studies reported that membrane resistance was increased after exposure to Aβ_1−42_ oligomers [54,55]. This enhanced R_m_, derived from differences in voltage response, could probably result from either TASK current mediated by TASK1 and TASK3—subtypes of leak K^+^ channels [56]—and/or subthreshold *I*_M_ and *I*_NaP_ currents [57]. Alternatively, because membrane resistance is strongly associated with neuronal morphology [58], it is also possible that the Aβ peptides also affected the biophysical properties of the membrane by altering membrane stiffness [59] and perturbing membrane permeability via both ion channel-dependent [51] and -independent mechanisms [60].

In addition, previous reports showing that Aβ_1-42_ oligomers induced hyperexcitability [30,54] and minimized rheobase to elicit AP [55] were consistent with our present data. Furthermore, our findings that Aβ_1-42_ oligomers did not affect other subthreshold membrane properties such as C_m_, Tau and RMP were supported by other studies [29,55,61].

Interestingly, neuronal hyperexcitability was not observed without the IA task, and Aβ_1-42_-induced hyperexcitability was specifically training-dependent. Contextual learning is accompanied by intracellular uptake of membrane proteins including receptors [62,63], and regulatory sites for voltage-gated sodium and potassium channels are located intracellularly [64,65]. These facts suggest that learning increases intracellular levels of Aβ_1-42_ [65,66] and that intracellular Aβ_1-42_ affects channel dynamics and the single-channel current of *I*_NaP_ [67], which may have decreased rheobase and increased excitability [68]. Since Nav1.6 channels were particularly expressed in CA1 pyramidal cells [65], it is possible that Nav1.6 is a target molecule of Aβ_1-42_.

To test our hypothesis, we applied RLZ, an antagonist of *I*_NaP_, to aCSF and found that Aβ_1-42_ oligomer-induced hyperexcitability was reduced by RLZ in a dose-dependent manner. In support of our data, another study showed that Aβ_1-42_-induced *I*_NaP_ enhancement, which mediates neuronal hyperexcitability, was reversed by RLZ [69]. Finally, our preliminary in vivo study showed that daily injection of RLZ into CA1 for 1 week ameliorated Aβ_1-42_ oligomer-induced impairment of contextual learning [70]. Taken together, these results suggest that Aβ_1-42_ oligomer-induced neuronal hyperexcitability with reduced rheobase may be mediated by *I*_NaP_ and that RLZ may have a protective effect against Aβ_1-42_-induced neuronal hyperexcitability.

### 4.3. Aβ_1-42_ Oligomers Selectively Impaired Other Hippocampus-Dependent Tasks

Since changes in sensory/motor or emotional functions may affect learning, basic physiological functions were assessed by using behavioral test batteries. We also confirmed that impaired contextual learning was not confounded by sensory/motor functions, pain sensitivity and emotional state. Consistent with our data, no detectable change in exploratory behavior and anxiety levels was reported in other studies [71,72].

Next, we observed that other hippocampus-dependent tasks were also selectively affected by Aβ_1-42_ oligomers, suggesting a specific impairment of hippocampal functions of dorsal CA1. The impairment of contextual freezing [40] and object recognition [71,72] was consistent with our present findings. With respect to social recognition memory, we observed that rats injected with Aβ_1-42_ oligomers were unable to discriminate between the novel and familiar social targets. Another study reported a delayed and persistent social memory deficit, although the social recognition memory impairment was not detected on day 7 post-injection [73].

Surprisingly, the intact working memory in our experiment was supported by another report showing that spontaneous alternation was not altered by Aβ_1-42_ oligomers [71], whereas many studies have shown that Aβ_1-42_ oligomers impaired spontaneous alternation in the Y-maze task probably due to variation in the target area and experimental paradigm [72,74,75]. Therefore, selective impairment of hippocampal-dependent tasks may be explained by a differential vulnerability of neural circuits as well as a specific contribution of Aβ_1-42_ oligomer-related pathology in the CA1 region.

### 4.4. Amyloid Deposition in the Target Area

Congo red staining, which detects amyloid aggregates in a compact rather than diffuse state, clearly showed amyloid deposition in CA1 pyramidal cells [47]. This was consistent with other reports showing Aβ deposition in CA1 pyramidal and DG granule neurons after injection of oligomeric Aβ_1-42_ [74,76]. Interestingly, we also found membrane labeling of CA1 pyramidal neurons using lecanemab, an anti-amyloid antibody specific for soluble Aβ_1-42_ species [77]. It is plausible that Aβ_1-42_ oligomers may have direct interactions with the cell membrane due to their sticky properties to facilitate adhesion [50], resulting in increased membrane resistance and immunoreactivity along the membrane. Two different techniques may suggest the presence of both β-pleated aggregates and soluble assemblies in the target area. It is likely that Aβ oligomers become insoluble after binding to the cell surface and may also undergo a transition between fibrils and lower-order species by polymerization or depolymerization [20].

Furthermore, lecanemab-positive non-pyramidal cells exhibited unique cell bodies with ramified branches, suggesting astrocytes and microglia [44,45] along the radial axis in the target area. Although we did not use glial cell-specific molecular markers, our current results were consistent with previous studies showing activated astrocytes and microglial cells surrounding and infiltrating amyloid deposits in the hippocampus [72,78]. Although analysis of Aβ_1-42_-induced neuronal cell death is also necessary for accurate analysis, our membrane labeling pattern suggests that the interaction between the oligomers and the cell membrane of CA1 pyramidal neurons may be responsible for the deleterious effects of Aβ_1-42_ on contextual learning and neuronal excitability.

## 5. Conclusions

Microinjection of Aβ_1-42_ oligomers into the CA1 region induced selective learning and memory deficits with concomitant neuronal hyperexcitability and amyloid deposition in the target area. Given the contextual learning-induced hyperexcitability of CA1 pyramidal neurons, the injection model is a useful technique for understanding the pathogenesis of AD and for testing drugs designed to protect against Aβ_1-42_ oligomers.

## Figures and Tables

**Figure 4 biomolecules-14-01425-f004:**
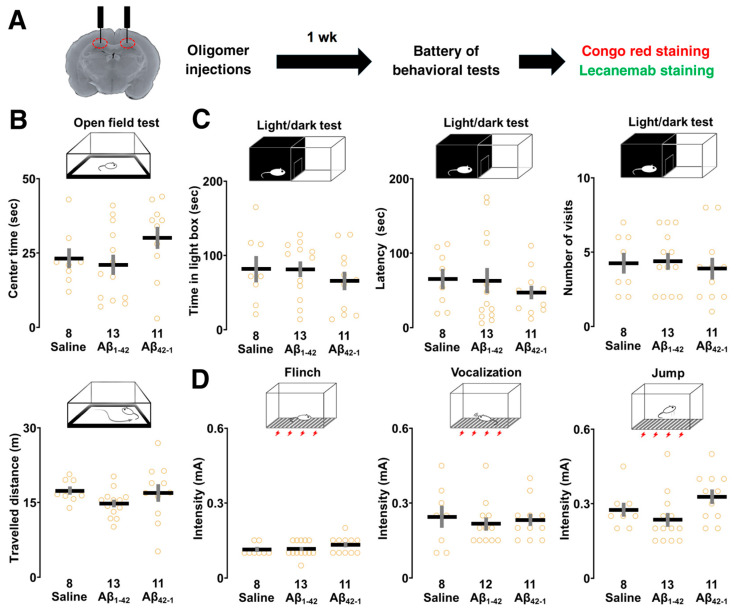
Effect of Aβ_1-42_ oligomers on the sensory/motor functions, pain sensitivity and emotional state. (**A**) Schematic illustration of the experimental design. Rats underwent the behavioral test battery 1 week after bilateral injections of Aβ_1-42_ oligomers into dorsal CA1 followed by Congo red and lecanemab staining. (**B**) The center time (upper panel) and traveled distance (lower panel) were not different in the open-field test. (**C**) The time spent in the lit chamber and the latency to enter as well as the number of visits in the light/dark test were similar. (**D**) In the flinch-jump test, the current thresholds for finch, vocalization and jump were not changed. The number of rats is shown at the bottom of each bar. Data were plotted as individual points and expressed as mean ± SEM.

**Figure 5 biomolecules-14-01425-f005:**
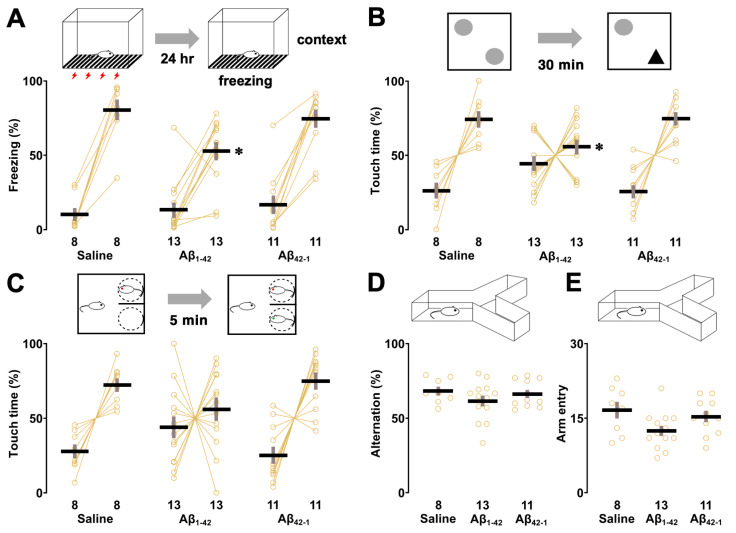
Effect of Aβ_1-42_ oligomers on other hippocampus-dependent tasks. (**A**) In the fear conditioning test, rats were allowed to explore the conditioning chamber and subjected to three electric foot shocks (0.8 mA, 2 s). After 24 h, re-exposure to the conditioning chamber increased freezing time, which was significantly reduced in Aβ_1-42_ oligomer-injected rats. (**B**) Exploration of the familiar and novel objects in the object recognition task. Training did not increase the touch time of the novel object in the Aβ_1-42_ oligomer-injected group. The exploration time of the novel object was also reduced in the Aβ_1-42_ oligomer group compared to the saline group. (**C**) Similarly, the social interaction time of both social targets was not different in the Aβ_1-42_ oligomer-injected group after training. The Aβ_1-42_ oligomer-injected group did not show a significantly longer touch time to the novel target. (**D**,**E**) The alternation ratio and the number of arm entries in the Y-maze test were not different. The number of rats is shown at the bottom of each bar. Data were plotted as individual points and expressed as mean ± SEM. * *p* < 0.05 vs. saline.

**Figure 6 biomolecules-14-01425-f006:**
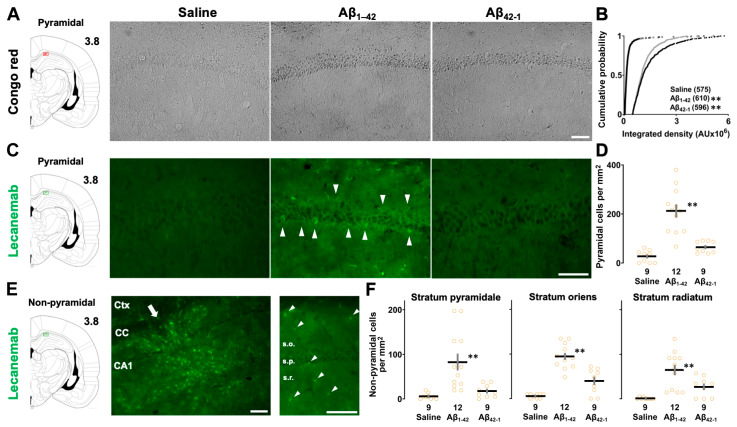
Quantification of amyloid deposition after 1 week of oligomer injection. (**A**) Diagram of coronal brain slice and representative micrographs of Congo red staining in the CA1 region. The number indicates the distance posterior to the bregma. (**B**) Cumulative distribution of staining density of individual CA1 pyramidal cells. The number of cells was shown in parentheses. AU, arbitrary unit. (**C**) Diagram of coronal brain section and representative fluorescence images of lecanemab immunoreactivity with arrowheads indicating pyramidal cells. (**D**) Semi-quantitative analysis of lecanemab-positive CA1 pyramidal cell counts per mm^2^. (**E**) Diagram of coronal brain slice and representative fluorescence images of lecanemab-positive non-pyramidal cells at the Aβ_1-42_ oligomer-injected site. The magnifications are ×100 (left) and ×200 (right), with the arrow indicating the injection track and the arrowheads indicating non-pyramidal cells. CA1, cornu ammonis 1; CC, corpus callosum; Ctx, cortex. (**F**) Semi-quantitative analysis of lecanemab-positive non-pyramidal cell counts per mm^2^ in the stratum pyramidale (s.p.), stratum oriens (s.o.) and stratum radiatum (s.r.) of the CA1 region. Scale bars: 100 μm. The number of sections is shown at the bottom of each bar. Data were plotted as individual points and expressed as mean ± SEM. ** *p* < 0.01 vs. saline.

## Data Availability

All data generated or analyzed during this study are included in this published article and its Appendix A.

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
