# Peer review of "Adverse Effects of Aβ1-42 Oligomers: Impaired Contextual Memory and Altered Intrinsic Properties of CA1 Pyramidal Neurons"

_biomolecules, 2024, doi:10.3390/biom14111425_

Round 1

Reviewer 1 Report

Comments and Suggestions for Authors

One of the hallmarks of AD is aggregation of beta-amyloid especially in its version 1-42. Whereas mature fibrils of beta-amyloid are less toxic the first species on aggregation pathways are oligomers which are the most toxic to neurons. Min-Kaung-Wint-Mon and colleagues venture deep into the mechanisms of this toxicity using solid  battery of behavioral tests, histochemical approach and electropysiological experiments. They found early effects of injections of oligomers into CA1 region of hippocampus which cause deficits in studied types of memory which they link to hyperexcitability of neuronal circuits in studied brain region. The paper is interesting due to showing early effects of amyloid oligomers injections, before the atrophy of brain occurs. However, some points need to be elucidated.

Major points: I wonder how different is the structure of control oligomers (beta amyloid 42-1) from the structure of beta amyloid 1-42 oligomers. The effects of these two do not differ much. Also it would be helpful if the authors could show these structures or refer to original papers with broader description.

It would help if there would be some control experiments checking neuronal death events.

I am not very familiar with behavioral experiments, therefore the other reviewer should pay special attention to these experiments and statistical interpretation of the results.

The paper is well written.

I recommend acceptance of the paper after major revision.

Author Response

Responses to the reviewers (biomolecules-3253629)

I would like to express my sincere gratitude for the reviewers’ helpful comments.

The specific responses to the reviewers’ comments were listed as follows:

---------------------------

Reviewers' comments:

---------------------------

One of the hallmarks of AD is aggregation of beta-amyloid especially in its version 1-42. Whereas mature fibrils of beta-amyloid are less toxic the first species on aggregation pathways are oligomers which are the most toxic to neurons. Min-Kaung-Wint-Mon and colleagues venture deep into the mechanisms of this toxicity using solid battery of behavioral tests, histochemical approach and electrophysiological experiments. They found early effects of injections of oligomers into CA1 region of hippocampus which cause deficits in studied types of memory which they link to hyperexcitability of neuronal circuits in studied brain region. The paper is interesting due to showing early effects of amyloid oligomers injections, before the atrophy of brain occurs. However, some points need to be elucidated.

Major points: I wonder how different is the structure of control oligomers (beta amyloid 42-1) from the structure of beta amyloid 1-42 oligomers. The effects of these two do not differ much. Also it would be helpful if the authors could show these structures or refer to original papers with broader description.

Response 1   Lines 728 - 739 and Figure S3

     Thank you for your insightful comment on the structure of the two oligomers. About the issue, we have added supplemental Figure S3 for the simulated structure of Aβ1-42 and Aβ42-1 peptides. Also, we reorganized a paragraph in Discussion with new references.

      Regarding the primary structure, Aβ1-42 peptide is segmented as N-terminal region (1-16), central hydrophobic cluster (17-21), turn A region (22-29), mid-hydrophobic region (30-35), turn B region (36-39) and C-terminal region at 40-42 [1] and the sequence is reverse in the Aβ42-1 peptide. The research on the reverse peptide is scarce, but we found only one publication with valuable information for clarifying your question. With the secondary structure, Aβ1-42 peptide undergoes gradual conformational change from a random coil to a β-sheet structure over the 24-hour of incubation whereas Aβ42-1 peptide demonstrates a strong β -sheet conformation almost immediately after preparation as evidenced by circular dichroism which was used to investigate the secondary structure of each peptide. Supporting the secondary structure, Aβ1-42 displays lag and elongation phases for fibrillogenesis but this pattern is not observed with the reverse peptide, Aβ42-1. Moreover, Aβ42-1 peptide forms fibrillar structures less ordered in appearance compared to Aβ1-42 peptide [2]. Consistent with the above findings, we found that Congo red-positive CA1 pyramidal neurons were more robust in the Aβ1-42-injected brains compared to the Aβ42-1-injected controls. We have revised a brief discussion about the structures of both oligomers related to their toxic effects. It would help if there would be some control experiments checking neuronal death events.

Response 2   Lines 822 - 823

     Thank you for your valuable comment on the important point. Due to the time constraint for revision (within 10 days), we could not be able to perform histological staining to check neuronal death. Instead, we described the limitation about the issue.

     In addition, we found several publications which could verify neuronal death. Consistent with our experimental timeline, 10 µM Aβ1-42 caused significant in vitro cell death of primary hippocampal neurons after 7 days of incubation whereas neuronal viability was slightly reduced by scramble Aβ but not affected by reverse control, Aβ42-1 [2]. On the other hand, in vivo infusion of 100 µM Aβ1-42 oligomers into the dentate gyrus triggered neuron loss of the target area after 1 week [3]. However, such in vivo neurotoxic effect was not observed in the dentate gyrus off both scramble [4] and reverse controls [5]. Therefore, neuronal death could probably contribute to memory deficits induced by Aβ1-42 oligomers whereas conclusion for Aβ42-1 is still limited.

I am not very familiar with behavioral experiments, therefore the other reviewer should pay special attention to these experiments and statistical interpretation of the results.

The paper is well written.

 I recommend acceptance of the paper after major revision.

Reference list for reviewer 1

  1. Urbanc, B., Cruz, L., Yun, S., Buldyrev, S. V., Bitan, G., Teplow, D. B., & Stanley, H. E. (2004). In silico study of amyloid β-protein folding and oligomerization. Proceedings of the National Academy of Sciences, 101(50), 17345–17350. https://doi.org/10.1073/pnas.0408153101.
  2. Vadukul, D. M., Gbajumo, O., Marshall, K. E., & Serpell, L. C. (2017). Amyloidogenicity and toxicity of the reverse and scrambled variants of amyloid‐β 1‐42. FEBS Letters, 591(5), 822–830. https://doi.org/10.1002/1873-3468.12590.
  3. Jean, Y. Y., Baleriola, J., Fà, M., Hengst, U., & Troy, C. M. (2015). Stereotaxic infusion of oligomeric amyloid-beta into the mouse hippocampus. Journal of Visualized Experiments, 100, 52805. https://doi.org/10.3791/52805.
  4. Brouillette, J., Caillierez, R., Zommer, N., Alves-Pires, C., Benilova, I., Blum, D., De Strooper, B., & Buee, L. (2012). Neurotoxicity and memory deficits induced by soluble low-molecular-weight amyloid-1-42 oligomers are revealed in vivo by using a novel animal model. Journal of Neuroscience, 32(23), 7852–7861.https://doi.org/10.1523/JNEUROSCI.5901-11.2012.
  5. Ryu, J. K., Franciosi, S., Sattayaprasert, P., Kim, S. U., & McLarnon, J. G. (2004). Minocycline inhibits neuronal death and glial activation induced by β‐amyloid peptide in rat hippocampus. Glia, 48(1), 85–90. https://doi.org/10.1002/glia.20051.

Reviewer 2 Report

Comments and Suggestions for Authors

The manuscript by Min-Kaung-Wint-Mon et al. describes changes to CA1 pyramidal neuron excitability in the presence of Aβ1-42 and Aβ42-1 synthetic oligomers compared to saline-treated controls. The article concluded that microinjection of the aforementioned synthesized oligomers increased neuronal excitability after a learning/training event, and that this effect could be reversed in electrophysiological recordings using a bath application of Riluzole, a glutamate modulator typically used in the treatment of ALS and Huntington’s Disease. Further, the authors found that oligomer-treated animals (whether they received the forward or reverse Aβ oligomer sequence) exhibited impaired performance on a battery of hippocampal-dependent behavioral tasks, such as fear conditioning and object recognition. The manuscript effectively tests the stated hypothesis and provides novel insight into the role of protein folding in Aβ plaque formation. General and specific comments follow below.

General Comments

The article is generally well structured, with few typographical/spelling errors. The references cited are appropriate in the context of the manuscript, including landmark studies and more recent literature.

The experimental design is appropriate to test the stated hypothesis, with one major caveat – the study only utilized male rats, without justification or explanation. It is hard to discern the reasoning as to why females were excluded. Further, it is well understood that the murine hippocampus is sexually dimorphic, meaning that the current study could be missing critical underlying sex differences in treatment/dose response.

The figures and graphs accurately depict the data. However, the authors did state multiple times in the manuscript that some data was not shown when supplementing that data could be helpful to readers and aid the overall interpretation of the manuscript’s results. Using only asterisks within the figures to denote significance would improve the figures, as the current “aa” or “a” indicator, in addition to the asterisks, make the figures cluttered and hard to interpret.

When reporting the statistics, the authors state, “…post hoc analysis revealed.... " However, the exact post hoc test(s) utilized in the one-way and two-way ANOVAs are not reported, only that the authors also used the Shapiro-Wilk test for normality. The post hoc test should be explicitly stated to aid in rigor and reproducibility.

Ethical considerations and data availability statements are adequate.

Specific Comments

The addition of a table or list of abbreviations would aid non-expert readers in comprehending the study’s findings. Many abbreviations are defined; however, this occurs in the article's methods section after many of these abbreviations have been presented to the reader.

The article could benefit from a small paragraph in the introduction regarding tauopathy in Alzheimer’s Disease (AD) and the synergy that tau and Aβ display in driving disease pathogenesis. While this article is specific to Aβ, it is vital to consider the larger picture and consider this as a potential study limitation, as we know that some AD patients can present with large amounts of amyloidosis post-mortem without any cognitive deficits. In contrast, others present with lower levels of amyloidosis but moderate to severe tauopathy and cognitive deficits. Although tau is not a target of the current study, the strength of the argument within this manuscript could be bolstered by briefly discussing tau.

The authors state little effort has been made to investigate the effects of Abeta1-42 oligomers on the intrinsic properties of CA1 pyramidal neurons. However, a search of the literature does produce over 100 results, so a more detailed discussion of the prior findings would be beneficial. 

It is assumed Abeta42-1 is selected as a control for Abeta1-42; however, the authors report effects with the oligomerized reverse peptide, and include some rationale for why that may be the case. An additional oligomerized peptide control would further support the findings. 

Validation of the oligomeric preparation would strengthen the rigor of the data (i.e. western blot of the Abeta preparation to view conformations).

While the number of cells recorded from were included, it is unclear how many different animals those cells came from. Please clarify. 

Comments on the Quality of English Language

The English is of sufficient quality. 

Author Response

Responses to the reviewers (biomolecules-3253629)

I would like to express my sincere gratitude for the reviewers’ helpful comments.

The specific responses to the reviewers’ comments were listed as follows:

---------------------------

Reviewers' comments:

---------------------------

The manuscript by Min-Kaung-Wint-Mon et al. describes changes to CA1 pyramidal neuron excitability in the presence of Aβ1-42 and Aβ42-1 synthetic oligomers compared to saline-treated controls. The article concluded that microinjection of the aforementioned synthesized oligomers increased neuronal excitability after a learning/training event, and that this effect could be reversed in electrophysiological recordings using a bath application of Riluzole, a glutamate modulator typically used in the treatment of ALS and Huntington’s Disease. Further, the authors found that oligomer-treated animals (whether they received the forward or reverse Aβ oligomer sequence) exhibited impaired performance on a battery of hippocampal-dependent behavioral tasks, such as fear conditioning and object recognition. The manuscript effectively tests the stated hypothesis and provides novel insight into the role of protein folding in Aβ plaque formation. General and specific comments follow below.

General Comments

The article is generally well structured, with few typographical/spelling errors. The references cited are appropriate in the context of the manuscript, including landmark studies and more recent literature.

The experimental design is appropriate to test the stated hypothesis, with one major caveat – the study only utilized male rats, without justification or explanation. It is hard to discern the reasoning as to why females were excluded. Further, it is well understood that the murine hippocampus is sexually dimorphic, meaning that the current study could be missing critical underlying sex differences in treatment/dose response.

Response 1   Lines 87 - 89.

     Thank you for your comment on the rationale of using only male rats in our study. Consistent with higher prevalence of AD in women compared to men [1], several transgenic female mice displayed more prominent amyloid pathologies, including Aβ deposition and cognitive decline, than their male counterparts [2, 3]. Interestingly, young females seem to be protected against Aβ toxicity by estrogen, whereas such protective effect diminishes when they approach menopausal age [4]. Moreover, estrogen has been shown to attenuate the neurotoxic effects of Aβ, including cognitive deficit, in the young ovariectomized 5xFAD mice [5]. Therefore, we rationalized to use only male rats in our study in order to circumvent the influence of gonadal hormones on the brain and Aβ pathology. We have revised our manuscript to explain why only male rats were included in our study.

The figures and graphs accurately depict the data. However, the authors did state multiple times in the manuscript that some data was not shown when supplementing that data could be helpful to readers and aid the overall interpretation of the manuscript’s results.

Response 2   Lines 775 - 777 and supplemental Figure S3

     Thank you for your suggestion. Due to the limitation of the number of figures, we decided to show the main data in the figures and to put the supportive data in the supplementary section. Now, we have uploaded those data with relevant statistical comparison in the supplementary file. Because the effect of RLZ in in vivo is particularly important for entire story of our experiment, we have added the reference.

Using only asterisks within the figures to denote significance would improve the figures, as the current “aa” or “a” indicator, in addition to the asterisks, make the figures cluttered and hard to interpret.

Response 3   All figures and the legends

     Thank you for your comment on the statistical indicator. We have deleted all “aa” or “a” indicator in all Figures and the legends.

When reporting the statistics, the authors state, “…post hoc analysis revealed.... " However, the exact post hoc test(s) utilized in the one-way and two-way ANOVAs are not reported, only that the authors also used the Shapiro-Wilk test for normality. The post hoc test should be explicitly stated to aid in rigor and reproducibility.

Response 4   Lines 352 - 355, 394 - 413, 463 - 467, 478 - 481, 578 - 581, 627 - 637, and 688 - 705.

      We described that Sidak’s multiple comparison test was used for post hoc analysis in the Materials and Methods section (line 337). In addition, we described the individual P values for the post hoc tests in Results section.

Ethical considerations and data availability statements are adequate.

 Specific Comments

The addition of a table or list of abbreviations would aid non-expert readers in comprehending the study’s findings. Many abbreviations are defined; however, this occurs in the article's methods section after many of these abbreviations have been presented to the reader.

Response 5   Lines 858 - 881.

     Thank you for the helpful advice. We have added an appendix of abbreviations after the Conclusion.

The article could benefit from a small paragraph in the introduction regarding tauopathy in Alzheimer’s Disease (AD) and the synergy that tau and Aβ display in driving disease pathogenesis. While this article is specific to Aβ, it is vital to consider the larger picture and consider this as a potential study limitation, as we know that some AD patients can present with large amounts of amyloidosis post-mortem without any cognitive deficits. In contrast, others present with lower levels of amyloidosis but moderate to severe tauopathy and cognitive deficits. Although tau is not a target of the current study, the strength of the argument within this manuscript could be bolstered by briefly discussing tau.

Response 6   Lines 35 - 42.

     Thank you for your insightful comment on the introduction. After critical literature review, we revised the introduction part in order to discuss the synergistic effects between Aβ and tau as well as to verify why Aβ is our target of study. Though our focus on Aβ could be a potential limitation, we also believe that our understanding based on previous research findings would align with the amyloid hypothesis that Aβ is the earlies trigger in initiating pathogenesis of Alzheimer's disease.

The authors state little effort has been made to investigate the effects of Abeta1-42 oligomers on the intrinsic properties of CA1 pyramidal neurons. However, a search of the literature does produce over 100 results, so a more detailed discussion of the prior findings would be beneficial.

Response 7   Lines 76 - 79.

     Thank you for your critical review of our statement. We did prior literature research before our study and noticed that there were a multitude of publications in which the researchers used hippocampal slices from transgenic [6] and naïve mice [7] without learning paradigm. Some bath-applied Aβ1-42 oligomers to the slice [8] and intracellularly in both in-vitro and in vivo conditions [9]. None of the data was obtained from the CA1 pyramidal neurons 1 week after direct in vivo injection of the oligomers into the target area. Moreover, neuronal hyperexcitability was evident after IA task, which implied that the effect of Aβ1-42 oligomers on intrinsic properties of CA1 was training-dependent. Therefore, we have revised the paragraph in the Introduction.

It is assumed Abeta42-1 is selected as a control for Abeta1-42; however, the authors report effects with the oligomerized reverse peptide, and include some rationale for why that may be the case. An additional oligomerized peptide control would further support the findings. 

Response 8   Lines 728 - 739 and supplemental Figure S3.

     We highly appreciate your suggestion concerning another oligomer control. However, due to the time constraint for revision (within 10 days), we could not be able to perform additional experiment with additional oligomerized peptide. Instead, we discussed about the molecular structure of Aβ1-42 and Aβ42-1 peptides in Discussion with the peptide analyses of Figures S2 and S3.

    To our knowledge, scramble Aβ has been used as a control in several papers. Yet, it shares some similar aggregation properties as Aβ1-42 peptide and triggers neurotoxic effect in the primary hippocampal neurons  [10]. Moreover, there are also sequence-specific effects in the Aβ1-42 peptide. For example, Aβ11–20, Aβ26–36, and Aβ31–42segments have high propensity for β-sheet formation and aggregation [11] and even a 3 amino-acid sequence (Aβ39-42) has been shown to modulate oligomerization of Aβ1-42 peptide [12]. Based on these findings, adequate knowledge of the amino acid sequence in the control peptide would help to facilitate better understanding of the toxic effects of Aβ1-42 oligomers as well as to circumvent sequence-specific effects of the control oligomers. Therefore, we assume that Aβ42-1 peptide would be a better fit for our study and are willing to get more information about the additional control.

Validation of the oligomeric preparation would strengthen the rigor of the data (i.e. western blot of the Abeta preparation to view conformations).

Response 9   Lines 100 - 102

     Thank you for pointing out the important data for validation. The co-author (Dr. Kimura) prepared the oligomers according to the published protocol which demonstrated formation of oligomeric assemblies of Aβ1-42 peptide [13]and added the reference. Using the oligomers, we have shown that Aβ1-42 oligomers induced LTP impairment in the hippocampal slices [14, 15]. Because we do realize the importance of validation of oligomers in our study, we would like to confirm this in our future publication, if you would permit.

While the number of cells recorded from were included, it is unclear how many different animals those cells came from. Please clarify. 

Response 10   Lines 89 - 94

     Thank you for your suggestion. We have added the number of rats for the current clamp data in the materials and methods section as follows: A total of 32 rats were used for behavioral analysis. A total of 15 rats were used for patch clamp analysis after IA training (Figure 2) and microinjected with saline (n = 7), Aβ1-42 (n = 4), or Aβ42-1 (n = 4). A total of 9 rats were used for patch clamp analysis without training (Figure S1) and microinjected with saline (n = 4) or Aβ1-42 (n = 5). In addition, a total of 6 rats were used for riluzole treatment (Figure 3) and microinjected with saline (n = 3) or Aβ1-42 (n = 3).

Reference list for reviewer 2

  1. 2022 Alzheimer’s disease facts and figures. (2022). Alzheimer’s & Dementia, 18(4), 700–789. https://doi.org/10.1002/alz.12638.
  2. Yang, J.-T., Wang, Z.-J., Cai, H.-Y., Yuan, L., Hu, M.-M., Wu, M.-N., & Qi, J.-S. (2018). Sex differences in neuropathology and cognitive behavior in APP/PS1/tau triple-transgenic mouse model of Alzheimer’s disease. Neuroscience Bulletin, 34(5), 736–746. https://doi.org/10.1007/s12264-018-0268-9.
  3. Dubal, D. B., Broestl, L., & Worden, K. (2012). Sex and gonadal hormones in mouse models of Alzheimer’s disease: What is relevant to the human condition? Biology of Sex Differences, 3(1), 24. https://doi.org/10.1186/2042-6410-3-24.
  4. Viña, J., & Lloret, A. (2010). Why women have more Alzheimer’s disease than men: gender and mitochondrial toxicity of amyloid-β peptide. Journal of Alzheimer’s Disease, 20(s2), S527–S533. https://doi.org/10.3233/JAD-2010-100501.
  5. Kim, J. Y., Mo, H., Kim, J., Kim, J. W., Nam, Y., Rim, Y. A., & Ju, J. H. (2022). Mitigating effect of estrogen in Alzheimer’s disease-mimicking cerebral organoid. Frontiers in Neuroscience, 16, 816174. https://doi.org/10.3389/fnins.2022.816174.
  6. Wang, X., Zhang, X.-G., Zhou, T.-T., Li, N., Jang, C.-Y., Xiao, Z.-C., Ma, Q.-H., & Li, S. (2016). Elevated neuronal excitability due to modulation of the voltage-gated sodium channel Nav1.6 by Aβ1−42. Frontiers in Neuroscience, 10. https://doi.org/10.3389/fnins.2016.00094.
  7. Eslamizade, M. J., Saffarzadeh, F., Mousavi, S. M. M., Meftahi, G. H., Hosseinmardi, N., Mehdizadeh, M., & Janahmadi, M. (2015). Alterations in CA1 pyramidal neuronal intrinsic excitability mediated by Ih channel currents in a rat model of amyloid beta pathology. Neuroscience, 305, 279–292. https://doi.org/10.1016/j.neuroscience.2015.07.087.
  8. Tamagnini, F., Scullion, S., Brown, J. T., & Randall, A. D. (2015). Intrinsic excitability changes induced by acute treatment of hippocampal CA1 pyramidal neurons with exogenous amyloid β peptide. Hippocampus, 25(7), 786–797. https://doi.org/10.1002/hipo.22403.
  9. Fernandez‐Perez, E. J., Muñoz, B., Bascuñan, D. A., Peters, C., Riffo‐Lepe, N. O., Espinoza, M. P., Morgan, P. J., Filippi, C., Bourboulou, R., Sengupta, U., Kayed, R., Epsztein, J., & Aguayo, L. G. (2021). Synaptic dysregulation and hyperexcitability induced by intracellular amyloid beta oligomers. Aging Cell, 20(9), e13455. https://doi.org/10.1111/acel.13455.
  10. Vadukul, D. M., Gbajumo, O., Marshall, K. E., & Serpell, L. C. (2017). Amyloidogenicity and toxicity of the reverse and scrambled variants of amyloid‐β 1‐42. FEBS Letters, 591(5), 822–830. https://doi.org/10.1002/1873-3468.12590.
  11. Abedin, F., Kandel, N., & Tatulian, S. A. (2021). Effects of Aβ-derived peptide fragments on fibrillogenesis of Aβ. Scientific Reports, 11(1), 19262. https://doi.org/10.1038/s41598-021-98644-y.
  12. Gessel, M. M., Wu, C., Li, H., Bitan, G., Shea, J.-E., & Bowers, M. T. (2012). Aβ(39–42) Modulates Aβ oligomerization but not fibril formation. Biochemistry, 51(1), 108–117. https://doi.org/10.1021/bi201520b.
  13. Stine, W. B., Dahlgren, K. N., Krafft, G. A., & LaDu, M. J. (2003). In Vitro Characterization of conditions for amyloid-β peptide oligomerization and fibrillogenesis. Journal of Biological Chemistry, 278(13), 11612–11622. https://doi.org/10.1074/jbc.M210207200.
  14. Kimura, R., MacTavish, D., Yang, J., Westaway, D., & Jhamandas, J. H. (2012). Beta amyloid-induced depression of hippocampal long-term potentiation is mediated through the amylin receptor. The Journal of Neuroscience, 32(48), 17401–17406. https://doi.org/10.1523/JNEUROSCI.3028-12.2012.
  15. Patel, A., Kimura, R., Fu, W., Soudy, R., MacTavish, D., Westaway, D., Yang, J., Davey, R. A., Zajac, J. D., & Jhamandas, J. H. (2021). Genetic depletion of amylin/calcitonin receptors improves memory and learning in transgenic Alzheimer’s disease mouse models. Molecular Neurobiology, 58(10), 5369–5382. https://doi.org/10.1007/s12035-021-02490-y.

Round 2

Reviewer 1 Report

Comments and Suggestions for Authors

I am satisfied with the changes in the manuscript